# CoDAT: Code maintenance, synthesis, and verification via LLM-based documentation

## Abstract

We present a a prototype tool, CoDAT, in which code can be linked to corresponding documentation, and the query "does the code correctly implemented the associated documentation"? can be posed to several large language models. The answers to the are either "yes, the code does correctly implement its documentation", or "no", the code does not correctly implement its documentation". These are displayed graphically, which enables a "birds eye view" in which the results of many queries can be viewed simultaneously. We show how CoDAT can be used to document, verify, and generate code.

**ACM Reference Format:**
Anonymous Author(s). 2026. CoDAT: Code maintenance, synthesis, and verification via LLM-based documentation. In . ACM, New York, NY, USA, 8 pages. https://doi.org/10.1145/nnnnnnn.nnnnnnn

## 1 Introduction

Recently, AI-based methods, and specifically Large Language Models (LLM), have been applied to automatically generate code from natural language queries. A key issue that has arisen, though, is the prevalence of a large number of errors in the resulting code, not all of which are corrected by subsequent manual code reviews. Consider the following quotes from the Sonar Code Report [20]:

> And while 82% of developers agree AI helps them code faster, and 71% say it helps solve complex problems more efficiently, this speed creates a new challenge: a trust gap. 96% of developers don't fully trust that AI-generated code is functionally correct.

and

> Given our finding in the prior section that 96% of developers have a hard time trusting that AI-generated code is functionally correct, you would think that verification of AI code is widespread. However, this is not the case: only 48% of developers always check their AI-assisted code before committing.

Since LLM's are trained using existing code, it is unsurprising that LLM-generated code contains errors, since existing manually written code does.

The above attitudes and practices on the part of developers highlight an urgent emerging problem: the introduction of large numbers of coding errors by the use of AI-generated code. When

such code is deployed in large infrastructure and/or safety critical applications, the consequences of failures due to code errors can be catastrophic. Such coding errors can also be exploited by bad actors to penetrate systems, e.g., for ransomware. Finally, safety-critical code must be understandable by humans: moral, professional, and legal liability concerns dictate that code upon which human lives rely cannot be left to unsupervised automation.

We propose to address the AI-coding problem by using large language models to check that code accurately implements functional specifications written in technical English, which we term the *code-documentation consistency* problem.

The importance and benefits of software documentation have long been recognized [2, 3, 15, 19]. Accurate, comprehensive, and clear software documentation helps maintain a "mental image" of the code, and therefore helps with the cognitive workload of:

- Figuring out how the code works,
- Tracing the code for particular scenarios, e.g., when fixing a bug, and
- How the core data structures are accessed and updated by the code. The data is the main object; the only purpose of code is to manage data.

Documentation therefore aids in code reviews, maintenance, and debugging. Unfortunately, it is also long recognized that maintaining up-to-date high-quality software documentation is expensive, detracts from the "primary task" of writing code, and is often neglected [3]. One proposed approach to the software documentation problem is to automate the generation of software documentation [18], e.g., by using learning, neural nets, etc...

We propose to build on and generalize earlier work by considering the problem of keeping the documentation up to date with the code, so that when code is changed (e.g., for debugging, maintenance, or upgrades), the documentation is updated accordingly. We formulate the *code-documentation consistency* problem as follows:

> *does the code correctly implement its associated documentation*?

We use this phrasing intentionally, rather than the logically equivalent: "does the documentation accurately describe the code?" The reason is one of emphasis: our preference is to write the documentation first, and then develop code based on the documentation.

The code-documentation consistency query admits a yes/no answer: either the code is correct w.r.t. the documentation, or it is not. This enables us to *use multiple LLM's* in concert: if all the LLM's certify the code as correct, we consider that the code is more likely to be correct than if a single LLM did so. Indeed, in experiments, we have observed that different LLM's sometimes give opposite answers to the code-documentation consistency query. This, in our opinion, is an argument for using multiple LLM's to increase confidence in the result. To date, LLM usage has been mostly for code generation. This limits the developer to a single LLM, as there is no straightforward way to combine the code generated by multiple

LLM's. Our use of LLM's to answer a yes/no query overcomes this problem, and avoids the weaknesses of any one particular LLM.

We have built a prototype implementation of the above ideas, as the CoDAT tool, which is a plugin for the Intellij IDE [1]. We now turn to the implications of our proposed work for code development, code reviews and code quality.

## 1.1 Code development

Code documentation must comprise the following: what does the code do and how does it do it? For a single method, the "what" is given by a *specification*, consisting of a *precondition* on the inputs to the method and a *postcondition* which relates the outputs to the inputs. The "how" is given by a *code sketch*, which outlines the key algorithmic ideas in the code, while abstracting away from coding details. The code sketch must be readable by itself, independently of the code that it describes. We also use *in-line code comments* in conjunction with code sketches, as discussed in detail below.

A key benefit of a well-written method specification is that a developer can invoke the method (e.g., as a "helper" method) without having to read the method's code to determine what the method does: reading the method specification is sufficient. In particular, the developer does not have to follow long method call chains, e.g., when method *A* calls method *B* which calls method *C* etc.. Thus, the specification serves as a *logical firewall* which separates the method invocation from the method implementation.

Now turning to the method implementation, we advocate that a good code sketch provides insight and continuity. The code sketch should be written first, rather than attempting to retroactively describe the code's purpose and functionality. The code sketch then guides the development and writing of the code itself, speeding up development while improving quality and reducing errors. A code sketch written before the code then pays off in reducing development time and improving code quality. A code sketch written after the fact is extra overhead. The writing of documentation as an *additional* activity to that of code development, rather than, as we propose, one that is *an integral part of code development*, is, in our opinion, a key reason that code documentation is often skipped, or not kept up to date as the code is debugged and updated [3].

## 1.2 Code Reviews

Code reviews are an important milestone in the life cycle of any major software project [14]. Traditional code reviews take a rigid and formal approach, focusing on depth and coverage rather than speed and scalability [12, 13]. Historically, code reviews have been effective in finding software bugs [6], [7], but, as code size has increased, the cost of code reviews has increased while their effectiveness has decreased.

Czerwonka et. al., [11] show that the traditional approach of formally reviewing code and providing feedback is negatively correlated with the size of the code review. The relatively high cost and formal requirements have lead to formulation and use of the *Modern Code Review* [8], which is lightweight, less formal, and tool-based. The use of code documentation and LLM-based code-consistency verification can contribute to reducing the costs and increasing the effectiveness of code reviews, since LLM's responses to the code-documentation consistency query serve to help find bugs in the code, in effect partially automating the debugging process.

Another important aspect to discuss about code reviews is the effect of code coverage on the code review's quality [17]. Maintaining up-to-date code documentation helps to improve the quality of code reviews. We will implement, in CoDAT, a change-tracking mechanism, so that when either code or its associated documentation is changed, the user is alerted to re-do the consistency checks. This is especially important for Modern Code Reviews, which are just-in-time, lightweight, and frequent. As a potential benefit, for example, code review comments and updates may be incorporated into inline comments to describe a particular change or implementation[19]. Yet, this effort may fall short if users are not routinely made aware of inline comments that need to be updated.

Emphasizing code coverage is an important link that can be easily maintained through just-in-time small but frequent code reviews. With this new focus on the small-scale approach to code reviews, up-to-date and accurate inline code comments are now more important than ever. For example, code review comments and updates may be incorporated into inline comments to describe a particular change or implementation[19]. Yet, this effort may fall short if users are not routinely made aware of inline comments that need to be updated.

## 1.3 Code Quality

We emphasize the following aspects of code quality: density of bugs (number of bugs per line of code) and readability. A code sketch summarizes, in natural language, the essential algorithmic information inherent in the code, while abstracting away from coding details. It is therefore easier to read than the code itself, while also focusing the reader on essential algorithmic information rather than accidental coding details (to use the terminology of Fred Brooks [9, 10]). Code sketches thus improve readability, which enhances the effectiveness of code reviews and code debugging, thereby improving code quality.

Our proposal to use multiple LLM's to check that code correctly implements the associated code sketch will also contribute to code quality by partially automating the code debugging process.

## 2 Our proposed approach to code documentation

We consider that intensive and well-maintained code documentation is crucial for software understandability, maintainability, and modifiability. Developers cannot write, maintain, and modify code with confidence and quality-assurance unless they have a good "mental picture" of how the code works. Unadorned code is notoriously difficult to understand, and so we advocate intensive code documentation at many levels of abstraction, so that documentation has a hierarchical structure. We propose the following documentation structure:

- *Functional specification of modules* i.e., what is a class or method required to do?
    - Classes: The purpose of the class and its major data structures (instance variables). Abstraction function and representation invariant.
    - Methods: a functional specification, given by two clauses:

```
//CS2.0:    If the map h for document d contains all the keywords
boolean b = true;
for(int i = 0; i < keys.length; i=i+1)
    b = b && h.containsKey(keys[i]);  //{b iff d contains all the keywords}
if (!b) return;   //Return with query unchanged

//CS2.1:        compute the total occurrence count sm for all keywords
int sm = 0;
for(int i = 0; i < keys.length; i=i+1)
    sm += ((Integer) h.get(keys[i])).intValue();

//CS2.2:        insert (d,sm) into the ArrayList of matches and then sort.
DocCnt dc = new DocCnt(d,sm);    //construct <d,sum>
matches.add(dc);                 //add it to matches
Collections.sort(matches, new SortByCount());  //re-sort matches
}
```

**Fig. 1: Highlighting of code-blocks in CoDAT**

(1) *Requires clause (precondition)*: constraints on inputs. The inputs of a method consist of its value parameters and the initial values of its reference parameters and instance variables (if any).

(2) *Effects clause (postcondition)*: states what the module does, as a relation between the outputs and the inputs. The outputs of a method consist of its return value and the final values of its reference parameters and instance variables (if any).

- *Code sketches*: how the code works, at a "high" level. A code sketch expresses the key algorithm underlying the code without getting bogged down in coding details. It therefore emphasizes the *essential algorithmic ideas* while abstracting away from the *accidental coding details* [9]. A code sketch is a list of *task-items*. Each task-item describes a single, coherent task that the code must accomplish. The code is a sequence of *task-blocks*; each task-block being a block of code which implements a single task-item. To aid readability, we place a copy of the task-item just before its corresponding task-block, as an in-line code comment.

- *In-line code comments*: more detailed description of the code, details of data structures and algorithms.

- *Refinement*: for complex task-items, we add detail with a "second-level" task-item that provides a more refined and operational description. This can be repeated until a level is reached that is straightforward to translate into working code. The number of levels needed depends on the difficulty of the implementation of the required functionality.

## 3 The CoDAT Tool

Our prototype implementation, the CoDAT tool, currently provides automated management of code documentation by augmenting the Intellij interface with the following information:

- *Type-tags*. Method headers and task-items have a type-tag placed in the vertical gutter next to the line numbers. For method headers this is an "m" inside a circle and for task-items this is an "f" inside a circle, the "f" denoting functionality. When the user left-clicks on the type-tag, a menu of available actions is shown. For example, selecting the "Run Analysis" menu items sends the code-documentation consistency query to all configured LLM's. The can be set using the "CoDAT Plugin Settings" item in the Intellij settings menu.

- *Status-tags*. For each task-item, status-tags are placed in the vertical gutter next to the line numbers, just to the right of the type-tag. There is one status-tag for each LLM in use. The status-tag has color that reflects the last yes/no answer that the corresponding LLM gave to the question: "does the task-block correctly implement the corresponding task-item"? Green indicates "yes" and red indicates "no".

- *Task-item labels*. A colored numeric label is placed immediately to the left of the task-item, e.g., CS0.1. This identifies the task-item uniquely within the code sketch. The task-item label maintains a summary of the LLM consistency queries. It is colored green if all LLM's answered "yes", e.g., CS0.1, red if all LLM's answered "no", e.g., CS0.1, and amber if the answers are a mix of "yes" and "no", e.g., CS0.1.

- *Code-documentation linkage*. Each task-item in the code sketch is linked to the corresponding task-block of code

```
 ⓜ                      void addDoc(Doc d, Map h) {  1 usage    ⚇ paulattie *
                            //REQUIRES: d is not null and h maps strings (the interesting words in d)
                            //    to integers (the occurrence count of the word in d)
                            //EFFECTS: If each keyword of this is in h, adds d to the matches of this.
                            //Code sketch:
 ⓕ Ⓜ Ⓒ Ⓣ                   //CS2.0:   If the map h for document d contains all the keywords,
 ⓕ Ⓜ Ⓒ Ⓣ                   //CS2.1:       compute the total occurrence count sm for all keywords
 ⓕ Ⓜ Ⓒ Ⓣ        💡         //CS2.2:       insert (d,sm) into the Arraylist of matches and then sort.

 ⓕ Ⓜ Ⓒ Ⓣ                   //CS2.0:   If the map h for document d contains all the keywords
                            boolean b = true;
                            for(int i = 0; i < keys.length; i=i+1)
                                b = b && h.containsKey(keys[i]);  //{b iff d contains all the keywords}
                            if (!b) return;   //Return with query unchanged

 ⓕ Ⓜ Ⓒ Ⓣ                   //CS2.1:       compute the total occurrence count sm for all keywords
                            int sm = 0;
                            for(int i = 0; i < keys.length; i=i+1)
                                sm += ((Integer) h.get(keys[i])).intValue();

 ⓕ Ⓜ Ⓒ Ⓣ                   //CS2.2:       insert (d,sm) into the Arraylist of matches and then sort.
                            DocCnt dc = new DocCnt(d,sm);    //construct <d,sum>
                            matches.add(dc);                 //add it to matches
                            Collections.sort(matches, new SortByCount());  //re-sort matches
                        }
```

Fig. 2: The addDoc method

```
 ⓕ Ⓖ Ⓒ Ⓣ                    //CS2.1:       compute the total occurrence count sum for all keywords
                            int sm = 0;
                            for(int i = 0; i < keys.length; i=i+1)
                                sm = ((Integer) h.get(keys[i])).intValue();
```

Fig. 3: Deliberately incorrect task block

```
 ⓕ Ⓜ Ⓒ Ⓣ                    //CS0.4:   Sort the matches
                            Collections.sort(matches, new SortByCount());
```

Fig. 4: Correct task-block

which implements the functionality described by that task-item. When the user left-clicks on the numeric label of the task-item, the corresponding task-block is highlighted, as shown in Figure 1, where the label CS2.1 has been clicked.

This graphical interface enables the developer to see the results of tens of LLM queries all at once. We believe that this will be of great help in using LLM's to analyze and develop code.

## 3.1 Example: document search engine

We have used CoDAT to analyze the document search engine program from [16, Chapter 13]. This program implements the search through a set of documents $d_0, \ldots, d_{n-1}$ using a list of search keywords $keys = w_0, \ldots, w_{k-1}$. The Query constructor creates a query with a single keyword $w$. It determines, for each document $d$, if $w$ occurs in $d$ at least once, in which case $d$ is a match. It then constructs a list of matching documents, ordered by the occurrence counts of $w$. There are also methods to add new keywords and new documents, which modify the existing query. Hence, at any time, there are documents $d_0, \ldots, d_{n-1}$ and keywords $w_0, \ldots, w_{k-1}$, which signify a query in progress. This query satisfies the following. For each matching document $d$, the *occurrence count* is given by the function $\mathsf{sumAll}(d, keys) = \mathsf{sumKey}(d, w_0) + \cdots + \mathsf{sumKey}(d, w_{k-1})$, where $\mathsf{sumKey}(d, w_j)$ is the number of times that $w_j$ occurs in document $d$. For a document to match, it must contain at least one occurrence of each keyword. The a list of matching documents is ordered by the occurrence counts.

Figure 2 shows the $\mathsf{addDoc}(Doc\ d, Map\ h)$ method. This method adds a document $d$ to the existing query in progress. It requires as input a hashmap $h$ which maps every "interesting" word in $d$ to its occurrence count in $d$. An interesting word is one which is not an article or a pronoun, such as "the", "she", etc. There are three task-items, CS2.0, CS2.1, CS2.2, that have corresponding task-blocks. Each task-block starts at its label and ends at the next label. The icons on the left, labeled with M, C, T, represent, the LLM's Mistral, Claude, and ChatGPT, respectively. The query that we submit to them is:

```
Determine whether or not the following
code correctly implements its associated
comment. Answer yes or no, as the first
word of your response. Then provide suggestions
for improving the code or comment, and give
any potential issues or best practices
that should be considered.
```

A green color indicates that the LLM answered "yes", and a red color that it answered "no". Since all results are mixed, the task-item labels are all colored amber. As we see, different LLM's disagree on the same queries. Note that the three task items all had different yes/no mixes. This justifies, in our opinion, our approach of using multiple LLM's.

Figure 3 shows the result when we deliberately introduce a bug: the line sm += ((Integer) h.get(keys[i])).intValue() is replaced by sm = ((Integer) h.get(keys[i])).intValue(), so that the occurrence count for keys[i] is not added to the total sm, but is assigned as its new value. All three LLM's correctly detected this error. Note that the task-item label CS2.1 is now colored red.

Figure 4 shows the very simple task-block which sorts the arraylist of matches. All the LLM's determined that the task-block is correct, and the task-item label CS0.4 is colored green. This task-block is from the Query constructor which creates the initial query with a single keyword.

## 4 Examples of mistakes by LLM's

We advocate the use of multiple LLM's because LLM's can make errors, even on relatively simple prompts, such as the following:

```
Please construct a red-black binary search
tree containing the following values: 15
8 2 3 10 7 18 25 8 15 10
```

Figure 5 shows a red-black binary search tree generated by Claude in response to this prompt. Claude first removed the duplicates, stating that: "Since red-black trees don't allow duplicates, I'll skip the duplicate values (8, 15, and 10 appear twice each)." This tree is incorrect, since the node with item 3 has, within it's left subtree, a node with larger item 7, which violates the binary search-tree requirement that all nodes in the left subtree of a node must have smaller or equal values. Figure 6 shows the tree generated by ChatGPT in response to this prompt. Instead of removing duplicates, ChatGPT added duplicate counts to the tree. The tree is correct. Figure 7 shows the tree generated by Mistral in response to this prompt. The tree is very incorrect, and appears to have elements of hallucination. These queries were executed on 31 January 2026 and 1 February 2026, so they represent the current state of the art, at the time of writing. This very wide variation in the results of a simple query (incorrect with one error, correct, and very incorrect) supports our position that the use of multiple LLM's will yield more accurate results than using a single LLM only.

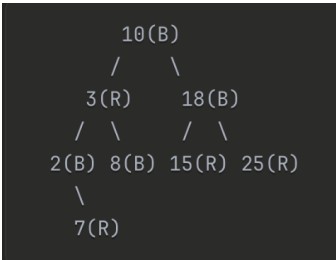

**Fig. 5: Incorrect red-black binary search tree generated by Claude**

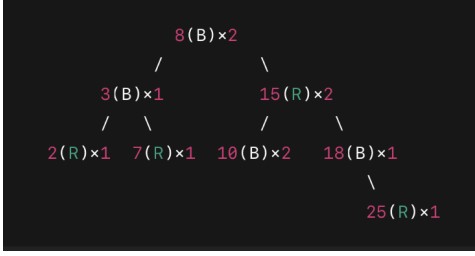

**Fig. 6: Incorrect red-black binary search tree generated by ChatGPT**

## 5 Prompt Engineering

Consider the addDoc method in Figure 2. We asked ChatGPT to generate code from the header:

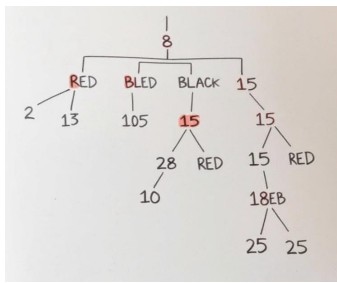

**Fig. 7: Incorrect red-black binary search tree generated by Mistral**

Given the following method header, specification
(REQUIRES and EFFECTS statements)
and Code sketch, please
generate code which correctly implements the code sketch

```
void addDoc(Doc d, Map h) {
  //REQUIRES: d is not null and h maps strings (the
       interesting words in d)
  //   to integers (the occurrence count of the word in d)
  //EFFECTS: If each keyword of this is in h, adds d to
       the matches of this.
  //Code sketch:
  //CS2.0: If the map h for document d contains all
       the keywords,
  //CS2.1: compute the total occurrence count sm for
  //        all keywords
  //CS2.2: insert (d,sm) into the Arraylist of matches and
  //       then sort.
```

ChatGPT generated code which was functionally identical to the code in Figure 2, with one notable exception. It produced

```
    sm += h.get(kw);
```

to add the next keyword count to sm. However, h has the raw type Map, and so h.get(kw) must first be cast to an Integer object, and then the value extracted using the intValue() instance method, as done in Figure 2. We then supplied CHatGPT with the declaration of the method that computes h, as in the following prompt:

The map h is actually produced by the following method
(which is in a different class).
Please adjust your code accordingly.

```
public Map addDoc(Doc d) {
  //REQUIRES: d is not null
  //MODIFIES:  this
 //EFFECTS: Adds all of the interesting words of d to this
 // with a count of their number of occurrences. Also
 // returns a hashtable (docMap) mapping each
 // interesting word in d to its number of occurrences.
  //HELPS: Engine.addDocs(u)
  //IMPL:
  //1. Compute the hashtable docMap and also the set of words in d
```

```
  //2. For each word w in d
  //    construct the DocCnt object dc = (d, docMap(w)) and
  //       insert the pair (w, dc) into this
  //3. Return docMap

  Map docMap = new HashMap();

  Iterator g = d.words(); //Generator for words in d's body
  Set docSet = new HashSet(); //Set to store all interesting
  //words in d's body,
  //to use later to iterate and insert into this

  String word;    //Used to store an individual word in
  //d's body that is being processed.

  //1.   Compute the hashtable docMap and also the set of
  //words in d (store in docSet)
  while (g.hasNext()) {
   word = (String) g.next(); //next word of d's body to process
   if (isInteresting(word)) { //if word is interesting, then

      //update its entry in docMap
      Integer ONE = new Integer(1);
      Integer count = (Integer) docMap.get(word);
      if (count == null) {
         count = ONE; }
      else {
         int value = count.intValue();
         count = new Integer(value + 1);
      }
      docMap.put(word, count);
      docSet.add(word);
      }
  }
  //docMap and docSet have the correct values:
  //    (A w : w in d.body and isInteresting(w) :
  //        docMap(w) = sumKey(s,w))
  //    docSet = {w : w in d.body and isInteresting(w)}

  //2. For each word w in d, i.e., in docSet
  //    construct the DocCnt object dc = (d, docMap(w)) and
  //       insert the pair (w, dc) into this
  Iterator gs = docSet.iterator();
  while (gs.hasNext()) { //iterate over words in docSet

    word = (String) gs.next();

   //Construct the DocCnt object dc = (d, docMap(word))
    int c = ((Integer) docMap.get(word)).intValue();
    DocCnt dc = new DocCnt(d, c);

    //Append dc to map(word)
    if (!map.containsKey(word))  { //word not in map,
                          //so map(word) is null
      ArrayList<DocCnt> A = new ArrayList<DocCnt>();
         //declare empty arraylist A
```

```
        A.add(dc);          //add dc to A
        map.put(word,A);  //insert (word,A) into map,
                          //i.e., into the wordTable
    }
    else { //word is in map
    ArrayList<DocCnt> A = (ArrayList<DocCnt>) map.get(word);
          //get map(word)
    A.add(dc);          //Append dc to it
    map.put(word,A); //Put updated ArrayList as new map(word)
    }

}
//Value of map is updated correctly
//    (A w : w in d.body and isInteresting(w) :
//        docMap(w) = sumKey(s,w))

    return docMap;

}
```

ChatGPT responded by making the necessary corrections, as follows:

```
Got it — h is the docMap returned by that other class's
addDoc(Doc d) method, and it's a raw Map whose keys are
interesting String words and whose values are Integer
counts.
So in addDoc(Doc d, Map h) we should:
Treat h as Map (raw), using casts like (Integer) h.get(kw).
Check keyword presence with h.containsKey(kw)
      (or h.get(kw) != null).
Sum counts via ((Integer) h.get(kw)).intValue().
```

As a result of adding the code of the addOdc method (of another class — wordTable) which generates the map cdh, the above line sm += h.get(kw); was changed to

```
    sm += ((Integer) h.get(kw)).intValue();
```

which is correct. The same queries submitted to Claude produced a similar result: the initial code contained the statement

```
sm += h.get(keyword);
```

for adding in the count for each keyword. After being given the text for the addDoc method which produces the map h, this was replaced by

```
Integer count = h.get(keyword);
sm += count.intValue();
```

Claude inferred that h.get(keyword); returns Integer objects, whereas ChatGPT used a cast to Integer.

This experiment shows that *context control* is crucial: how much of the surrounding code should be included in a prompt? It shows that adding some context can correct errors in the generated code. However, including too much context will slow down the prompt processing, and may in fact be counter productive, leading to less accurate code.

CoDAT is an ideal platform for conducting experiments to answer this question, and similar ones.

## 6 Future work

We are working to extend the functionality of CoDAT in several directions. Key to this effort is the execution of large ambitious case studies, the feedback from which will generate ideas for functional improvements for CoDAT. CoDAT is easy to use, with a minimal learning curve, and should be deployable in undergraduate and graduate programming and software engineering courses.

- *Software correctness.* We will use CoDAT to generate Hoare logic proof scripts that are then proof-checked using tools such as Key-Hoare [4]. We will conduct experiments to determine if using CoDAT makes formal code correctness proofs less costly.
- *Code reviews.* We will use CoDAT to perform code reviews, and will compare the results with code review studies from the literature.
- *Software security.* We will use CoDAT to analyze software for security vulnerabilities, and to effect necessary fixes. We will then subject the software to "red teams" who will attempt to penetrate the software, i.e., "capture the flag". This will evaluate the effectiveness of CoDAT in securing software. We have already used LLM's to discover and report eighteen assigned CVEs affecting major vendors including Apple, Adobe, Shopify, and WordPress. These vulnerabilities span multiple weakness classes, including memory safety errors, improper input validation, command injection, cross-site scripting, cross-site request forgery, and information exposure. These are not listed in this submission to preserve author anonymity.
- *Software safety.* Software safety is the resilience of software to failures resulting from any number of faults, including software bugs, human error, physical component failure, etc.. Thus, safety goes beyond functional correctness, and requires an analysis of the interaction between a system and its environment, e.g., using game-theoretic semantics [5].
  We are currently working on modeling a system for nuclear waste management, as a case study. We will test its safety and robustness by injecting faults, which model environment faults, human operator error, physical component failure etc.
- *Legacy software.* We will select a major piece of legacy code, and will use CoDAT to generate documentation and verify the code w.r.t. the documentation.

## 7 Conclusions

The benefits of the effective use of large language models in system development cannot be overstated. Large language models are already in very widespread use for coding, and this use is a significant component of the impact of Artificial Intelligence on the software industry, both in terms of the economics and also the practice of system development. It follows that any improved methodologies and techniques for the use of LLM's in system development will have outsize impact.

We believe that the CoDAT project will yield benefits that include increased code productivity together with higher quality code that

is both more correct (contains fewer errors) and more understandable (has better documentation). Prompt engineering, the use of multiple LLM's, the graphical interface, and large case studies, will be the focus of our work going forwards.

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
