# OpenReview forum: "CoDAT: Code maintenance, synthesis, and verification via LLM-based documentation"
_ACM.org/AIWare/2026/Conference — Submitted to AIware 2026_

### Official Review · Reviewer_MB7t · 2026-03-07

**Rating:** 1
**Confidence:** 5

**Review:**

Strengths:

This tool is relevant to the software engineering community as it allows for keeping documentation up-to-date.

Weaknesses:

There is no comparison with similar tools.

The article does not present metrics for evaluation.

The article presents only one use case and therefore does not offer evidence of positive results for a variety of situations.

Comments:

The article is easy to understand. However, some sentences need references.

The figures could be closer to the corresponding text. Also, when images contain many details, an index would indicate the correct location in the image description. The code generated in section 5 would be part of a replication package. The authors could have highlighted specific lines of the methods to help the reader understand the process. The article presented the result of a single sample, which is not sufficient to suggest that the implementation is capable of handling various errors.

No specific metrics were used to measure the success of CoDAT compared to similar tools.

The article did not provide details on the use of prompts, strategies, or adjusted parameters.
Sovrano, Francesco, et al. "Simplifying software compliance: AI technologies in drafting technical documentation for the AI Act." Empirical Software Engineering 30.4 (2025): 91.

To date, it is unclear to what extent CoDAT differs from other automated documentation tools.

The tool prototype is not available for evaluation.

Tools that provide automated code verification with documentation can improve the quality of code generated by humans or AI and are highly relevant for Software Engineers.

implementation[19]. -> missing white space

**Summary:**

This article presents CoDAT, a code checker based on pre-existing documentation. CoDAT uses multiple LLMs (Leadership Learning Methods) to verify if the generated code conforms to existing documentation. The tool identifies documentation compliance through the application of three LLMs. An example with CoDAT results is provided.

---

### Official Review · Reviewer_8qmY · 2026-03-09

**Rating:** 2
**Confidence:** 5

**Review:**

Strengths:
The paper addresses an important and timely problem: the growing reliance on LLM-generated code and the associated trust and correctness concerns. The idea of verifying code against documentation using LLMs is interesting and aligns well with the goals of AI-powered software development. The paper also emphasizes documentation-driven development and proposes a practical IDE integration, which could potentially help developers maintain consistency between specifications and implementation. The use of multiple LLMs to mitigate the unreliability of individual models is a sensible design choice. The paper includes illustrative examples showing how LLMs may disagree and how the tool can highlight potential implementation errors.

Weaknesses
Writing quality and clarity.
The paper requires substantial improvement in writing quality and proofreading. Several sentences contain grammatical errors or unclear phrasing. For example, the sentence “does the code correctly implemented the associated documentation?” should be rewritten as “does the code correctly implement the associated documentation?”. Similarly, the sentence “The answers to the are either…” should be revised to “The answers can be either…. Or etc... needs to be etc.
but, as code size has increased, the cost of code reviews has increased while their effectiveness has decreased [need a reference here]

Software engineering already provides well-established documentation standards and practices (e.g., inline documentation, API documentation, design-by-contract, formal specifications). The paper proposes a new structured documentation approach but does not sufficiently explain why existing documentations templated proposed by SE standards are inadequate. The authors should clearly discuss the limitations of current documentation practices and provide empirical evidence supporting the need for a new format.

The paper mentions that CoDAT can be used for code generation, but this functionality is not clearly explained. It is unclear how code should be generated from the documentation or whether CoDAT actually supports this capability. The description suggests that the tool primarily performs verification rather than generation. This aspect needs clarification.

The paper focuses mainly on verifying whether code conforms to documentation. However, the approach appears to be inherently bidirectional, as it can also identify cases where the documentation does not accurately reflect the implementation. This important aspect is not clearly stated in the abstract or emphasized in the paper and should be highlighted.

The current title does not clearly communicate the core idea of the work. A title that explicitly reflects the LLM-based verification aspect might be more appropriate. For example:
“CoDAT: Code Maintenance, Synthesis, and Verification via LLM-based Documentation Consistency Checking.”

The evaluation mainly consists of illustrative examples rather than a systematic empirical study. The paper does not present quantitative evidence demonstrating improvements in bug detection, developer productivity, or documentation quality. A stronger empirical evaluation on real-world codebases would significantly strengthen the work.

The approach assumes that developers will write detailed, structured documentation before implementing code. In practice, such documentation is often missing or incomplete in many software projects. The paper should discuss the practicality of this assumption and how the approach would apply to existing systems. One justification is comparing the required effort to write this heavy document against the prompts for the code fix-generation ,... It is better to split the objectives and evaluate them one by one.

Using multi-LLMs for this setup is not in fact the contribution of the paper as it has been studied and discussed in any LLM-as-judge approches. Aothors however, need to discuss and evaluate some hyperparameters of these models, such as temperature. Also the majority vote or LLM-as-judge concept needs to be mentioned as a part of the literature.

- Scalability in terms of execution time is also interesting to study.

**Summary:**

The paper introduces CoDAT, a prototype IDE plugin that links structured documentation with code and uses multiple large language models (LLMs) to check whether the code correctly implements the documentation. The approach relies on documentation written in a structured format (e.g., specifications and code sketches). The tool sends consistency queries to multiple LLMs and aggregates their yes/no responses to help detect bugs, maintain documentation, and assist developers during code development and review.

---

### Official Review · Reviewer_LcP8 · 2026-03-10

**Rating:** 1
**Confidence:** 3

**Review:**

The working prototype is a real contribution, but the submission in its current form is not yet ready for publication. The primary concerns are about presentation and rigor rather than the underlying ideas.

**Writing quality**
The manuscript contains numerous typographical errors, duplicate paragraphs (section 1.2 contains a near-verbatim repetition spanning roughly half the section), grammatical mistakes, and at least one inconsistency between a figure caption and the main text (Figure 6 is labeled 'incorrect' while the body describes the same tree as correct).
This level of proofreading is below the standard expected at the venue and suggests the paper may have been submitted prematurely.

**Experimental rigor**
The evaluation consists of what looks like a single case study on one textbook program. Claims about the benefits of using multiple LLMs are illustrated with three queries rather than any systematic analysis. Without even a modest controlled experiment, it is difficult to assess whether the approach leads to reliable or reproducible results.

**Contribution boundary**
The paper would benefit from a sharper related work section and a more explicit statement of claims, situating CoDAT relative to existing work on LLM-based code review and documentation generation.

I would encourage the authors to develop this work further before resubmission - the core idea of fine-grained code-documentation linkage with LLM verification & a tool automating it is worth pursuing but the current draft requires substantial revision.

**Summary:**

The paper addresses a genuinely important practical problem - using LLMs to verify code against structured documentation and demonstrates a working prototype integrating with actual developer tools (IDE plugin) along with some promising early results.